# Synthesis and Characterization of 𝒌-Carrageenan/PVA Nanocomposite Hydrogels in Combination with MgZnO Nanoparticles to Evaluate the Catechin Release

**DOI:** 10.3390/polym15020272

**Published:** 2023-01-05

**Authors:** Farzaneh Sabbagh, Nadia Mahmoudi Khatir, Khadijeh Kiarostami

**Affiliations:** 1Department of Botany and Plant Science, Faculty of Biological Science, Alzahra University, Tehran 1993891176, Iran; 2Department of Chemical Engineering, Chungbuk National University, Cheongju 28644, Republic of Korea; 3Department of Biotechnology, Faculty of Biological Science, Alzahra University, Tehran 1993891176, Iran

**Keywords:** polyvinyl alcohol, 𝒌-carrageenan, sol–gel, drug release, nanoparticle

## Abstract

In the current study, nanocomposites were prepared by combining k-carrageenan, polyvinyl alcohol (PVA), and doped nanoparticles (Magnesium oxide) MgO, (Magnesium Zinc oxide) MgZnO 1%, MgZnO 3%, and MgZnO 5%. The nanoparticles were synthesized by a sol–gel method and mixed with a mixture of k-carrageenan/PVA (Ca/PVA) in various ratios. The structure of the composites was analyzed using thermogravimetric analysis (TGA), X-ray diffraction (XRD), and Fourier transform infrared spectroscopy (FTIR). The Ca/PVA mixture was then mixed with nanoparticles and loaded with active ingredient, catechin. Scanning electron microscope (SEM) and texture analysis were performed to analyze the nanocomposites. Entrapment efficiency (EE%) and drug release studies confirmed that k-carrageenan/PVA/MgZnO 5% had the highest EE% at 81.58% and a drug release of 75.21% ± 0.94. The EE% of k-carrageenan/PVA/MgO was 55.21% and its drug release was 45%. This indicates that ZnO plays an effective role in the structure and performance of Ca/PVA composites. The SEM images of MgO composites show smoother surfaces compared to MgZnO composites. This may be one of the reasons for the increased EE% and drug release of MgZnO composites. The addition of ZnO to the composite structure can lead to the appearance of pores on the surface of the composite, increasing entrapment and drug release.

## 1. Introduction

The interesting effects of tea flavonoids such as catechins have been demonstrated by various researchers over the years. Catechin is a natural component normally found in tea leaves. Based on evidence from various reports, catechins have significant antidiabetic [1], antioxidant, anti-HIV [2], anti-obesity [3], anti-inflammatory [4], anti-aging [5], antibacterial [6], hypolipidemic [7], and neuroprotective [8] effects on the body [9]. Green tea polyphenols have been used as food additives to enhance the shelf-life and antioxidant properties of foods [10]. By the oral administration of these compounds, measurements of pharmacokinetic parameters showed the low oral bioavailability of catechins (<5%), a high first-pass effect, and poor intestinal absorption [11]. Despite historical therapeutic success, there are several challenges in the delivery of catechins, including poor targeting and poor bioavailability [12]. This leads to only a small percentage of the administered catechin reaching the target organ, with a greater amount being distributed to other parts of the body [13]. Recently, encapsulated nanoscale carriers have attracted attention to enhance the therapeutic efficacy of therapeutic compositions such as drugs and vitamins [14]. By encapsulating functional nanoparticles in a matrix, it is also possible to stimulate porous hydrogel systems to release therapeutic complexes. Nanoparticle encapsulation in hydrogels is similar to drug encapsulation [15]. Nanocomposites can offer significant advantages such as improved membrane permeability, ease of fabrication, and high drug solubility in therapeutic formulations [16]. Several nanoparticle delivery systems have been tested to overcome the problems of oral delivery [11]. The characteristics of metal oxide nanoparticles that make them ideal candidates for biomedical applications stems from unique specifications such as high surface response and absorption capacity. Zinc oxide (ZnO) is one of the most favorable metal nanoparticles due to its thermal and chemical stability, non-toxicity, biocompatibility, and low cost.

Polyvinyl alcohol (PVA) has attracted the attention of scientists as a non-ionic and water-soluble substance. Due to its physical gelling properties, non-toxicity, and biocompatibility, this polymer has been widely applied in the field of drug delivery. PVA-based hydrogels have shown some limitations due to their lack of ionic appendages and low degree of swelling. Introducing ionic biopolymers into these hydrogels represents an efficient approach to improve the ionic properties of PVA hydrogels. k-carrageenan is an ionic biopolymer with active functional groups that is applied in the synthesis of PVA-based hydrogels in this study. Although polysaccharide-based films exhibit most of the desirable properties for wound dressing applications, they also have limitations such as poor chemical stability in water, relatively low mechanical strength, and low water permeability. Therefore, there is a need to improve these properties of films to improve their performance in wound healing applications. By combining PVA and k-carrageenan, it is possible to make a dressing material for wound healing. We propose a Ca/PVA composite hydrogel. A ratio of k-carrageenan to PVA was prepared and characterized by FTIR, XRD, and TGA. The selected Ca/PVA composites were then mixed with nanoparticles of MgZnO with three different ZnO ratios. A composite loaded with MgO was chosen as the control composite. All nanocomposites were examined with textural analysis and microstructural studies. Catechins were loaded into the nanocomposites and the drug loading, entrapment efficiency, and in vitro drug release of catechins from Ca/PVA/MgZnO nanocomposites were evaluated. 

## 2. Materials and Methods

k-carrageenan, gelatin, polyvinyl alcohol (PVA) (MW~93.5000), (+)-catechin hydrate, zinc nitrate hexahydrate, magnesium nitrate hexahydrate, potassium dihydrogen phosphate, potassium chloride disodium hydrogen phosphate, and sodium chloride (pH 7.4, 135 mM sodium, 10 mM phosphate, 2.7 mM potassium) were used to prepare the phosphate buffer saline (PBS) used in this study and was obtained from Sigma-Aldrich (Jefferson City, MS, USA) and used as received. (+)-catechin hydrate standard solution for high pressure liquid chromatography (HPLC) (YL 9100, Younglin Inc., Anyang, Republic of Korea) was purchased from TCI Japan with a purity greater than 97.0%. Distilled water was used for the preparation of all solutions.

### 2.1. 𝑘-Carrageenan/PVA Composite Production

Nanocomposites were prepared in situ by mixing of PVA and k-carrageenan polymer solutions. To prepare the hydrogels, each polymer was prepared separately and both polymers were mixed in the liquid state and stirred for 1 h until a homogenous, transparent, and viscous solution without air bubbles was obtained. The required initial compositions for hydrogel preparation are presented in Table 1. For the preparation of the PVA polymer solution, an amount of 1 g of PVA was dissolved in 20 mL distilled water at 80 °C. To prepare the k-carrageenan solution, 1 g of k-carrageenan was dissolved in 25 mL distilled water at 70 °C. Both solutions were stirred until completely clear [17].

Finally, 0.8 mg/mL aqueous solutions of MgO, MgZnO 1%, MgZnO 3%, and MgZnO 5% nanoparticles were prepared according to [18]. MgO nanocomposites were evaluated as the control hydrogel.

### 2.2. Measurements

A mixture containing approximately 2 mg of each A–D hydrogel sample in about 100 mg potassium bromide (KBr) was ground using a pestle and mortar on a ZnSe plate using a high-pressure clamp. Fourier-transform infrared spectroscopy (FTIR) spectra were collected (Bruker Vertex 70, Minneapolis, MN, USA) at wavelengths of 400–4000 cm^−1^. Sixteen scans were accumulated to obtain a reasonable signal-to-noise ratio.

X-ray diffraction (XRD) analyses were performed on a Philips X’PERT instrument at room temperature. The powdered sample was placed in a plastic sample holder and the surface was smoothed with a glass slide. All A–D hydrogels were analyzed using XRD with a powder diffractometer that relies on Cu Kα radiation (λ = 1.54 Å).

Thermogravimetric analysis (TGA) was performed using a thermal gravimetric analyzer (Shimadzu TGA-50, Kyoto, Japan). The TGA of all A–D hydrogels was analyzed using a Shimadzu TGA-50 on 4 mg sample. Nitrogen gas was used as the heating medium at a flow rate of 20 mL/min. The composite was heated at a rate of 10 °C/min at temperatures ranging from 30–800 °C.

### 2.3. Microstructure Studies

The morphology of the nanocomposites was analyzed using scanning electron microscopy (SEM, LEO-1530, Zeiss, Jena, Germany). Nanocomposites were made by applying and coating a small number of dry nanocomposites on carbon tapes. Fully dried hydrogel discs were cut to an optimal size and were attached to double-sided tape placed on to aluminum stubs. The stubs were coated with ~300 Å-thick gold under an argon atmosphere using a gold sputtering module in a high-vacuum evaporator [19].

### 2.4. Texture Analysis

The adhesiveness, hardness, consistency, and springiness of the nanocomposites were determined using a texture analyzer (TA. XT, London, UK). The diagnostic probe P/0.4 (stainless steel cylinder with a diameter of 4 mm) was squeezed at 1.0 mm/s into the nanocomposites (20 mm height), with a release force of 3 g, in triplicate, and to a sample depth of 70% [20].

### 2.5. Catechin Encapsulation

For catechin encapsulation into nanocomposites, prepared composites were immersed in 5 mg/mL catechin solution and kept at 4 °C for 48 h. To remove the additional catechin on the surface and to minimize eruption release experiments, after encapsulating the catechins, the nanocomposites were soaked in distilled water for approximately 5 min [21].

### 2.6. Study of Entrapment Efficiency and Drug Loading of the Nanocomposites

Hydrogel samples of specified weight (W_i_) containing a specified amount (W_0_) of catechin drug were swollen in 100 mL of water prior to drug loading. The drug-loaded wet hydrogel samples were carefully removed from the aqueous buffer solution after 48 h of swelling and washed with the same solution to remove unbound drug from the samples. Using equations 1 and 2, the drugs loading (DL) and entrapment efficiency of the nanocomposites were calculated as follows:DL (mg/g hydrogel sample) = Wd − Wi/Wi(1)
Entrapment efficiency (%) = Wd − Wi/W0 × 100(2)
where Wd is weight of the drug-loaded dry hydrogel sample [22].

### 2.7. Release Studies in Phosphate-Buffered Saline (PBS) Buffer

In vitro release studies of catechins from nanocomposites were carried out at 37 ± 0.5 °C and a rotation speed of 50 rpm in 20 mL of PBS solution (pH 7.5) for 24 h. All the catechin-loaded nanocomposites were then immersed in PBS media separately. At several time intervals, 1 mL of the solution containing the released drug was withdrawn and at the same time, to keep the solution volume constant, 1 mL of fresh solution was added. The concentration of released catechins in the drawn solution was analyzed with a UV–Vis spectrophotometer at 228 nm. All release experiments were carried out in triplicate and average values were considered.

### 2.8. Cell Toxicity Studies

To test the cytotoxicity of the film, a L929 mouse fibroblast cell line was used. For this, the cryopreserved L929 cells were thawed at 37 °C and cultured in Dulbecco’s modified Eagle medium (DMEM), containing 4.5 g/L glucose, 10% fetal bovine serum, L-glutamine, 3.7 g/L NaHCO3, sodium pyruvate, and 1% antimycotic solution under 5% CO_2_ at 37 °C and 95% humidified air. The medium was replaced every 2 days and the cells were passaged at 85% confluence. Up to 5 passaged cells were used for the test. The cell viability of fibroblasts was tested by the MTT method using 3-(4,5-dimethylthiazol-2-yl)-2,5-diphenyl tetrazolium bromide (MTT). For this, film disks (8 mm in diameter) were placed in 24-well plates, sterilized with 75% ethanol for 30 min, irradiated with UV for 30 min, and washed twice with phosphate-buffered saline (PBS). A suspension of L929 cells (2 × 10^4^ cell/mL) was inoculated into each well and cultured for 24 h, 48 h, and 72 h. Then, the media was aspirated and 300 μL of MTT solution was added to each well and incubated at 37 °C for 4 h. The media was then removed, was added 300 μL of DMSO, mixed well using a shaker for 5 min, and then transferred to a 96-well plate to measure the absorbance at 570 nm using a microplate reader.

### 2.9. Statistical Analysis

The quantitative comparison of drug release under the influence of different formulations was compared using one-way ANOVA. A *p* < 0.05 was considered significant. The statistical analysis values are presented as mean ± standard deviation.

## 3. Results

### 3.1. FTIR and XRD

Figure 1a shows the FTIR spectra of all k-carrageenan/PVA formulations. For simplicity, we will refer to the samples as A, B, C, and D, respectively. The spectra of all samples indicated the existence of O–H stretching vibrations, indicated by an extensive absorption at 3400 cm^−1^. The peak at 2940 cm^−1^ in all the Ca/P formulations is responsible for the presence of –OH stretching groups in the carrageenan structure. The band range of 1700–1725 cm^−1^ corresponds to carboxylic acid (1699 cm^−1^ identifies carboxylic acid groups) and the broad band at 1633 cm^−1^ corresponds to C=C stretching vibrations. The peak detected at 3200 cm^−1^ correlates with O–H stretching for the inter- and intra-molecular hydrogen bonds. Between 2840 and 3000 cm^−1^, there is a vibration band associated with the C–H stretching of alkyl groups. The complex is observed to have sharp absorption peaks at 1259 and 1249 cm^−1^, because of the S=O bond of sulfate esters. The band at 845 cm^−1^ is attributed to D-galactose-4-sulfate, and the fairly strong band at around 930 cm^−1^ indicates the presence of 3,6-anhydro-d-galactose (DA). Furthermore, the peak at approximately 805 cm^−1^ indicates the presence of a sulfate ester in the anhydro-d-galactose residue, a carrageenan-specific band. The FTIR spectra of P/Ca hydrogels have been obtained by other authors. Bajpai and Daheria (2014) [22] prepared k-carrageenan and PVA using glutaraldehyde as a crosslinker. The crosslinking reaction between PVA and carrageenan was confirmed by FTIR analysis.

The XRD diffraction patterns of Ca and PVA of different percentages are shown in Figure 1b. XRD patterns for Ca 100%, Ca/PVA 50:50, Ca/PVA 60:40, and Ca/PVA 70:30 presented typical crystalline peaks at about 21.58°, 21.38°, 21.78°, and 21.28°, respectively. Ca has an amorphous nature and PVA has a semi-crystalline state. In the Ca/PVA 50:50 and Ca/PVA 60:40 combinations, there are no significant differences between peaks, but in the Ca/PVA 70:30 combination, the percentage reduction the peak intensities are highlighted.

### 3.2. TGA

TGA thermograms of Ca 100%, Ca/PVA 50:50, Ca/PVA 60:40, and Ca/PVA 70:30 hydrogel were studied, and Table 2 displays the results. The thermal treatment was started at 30℃ and raised to 800 ℃ at a constant rate of 10 ℃/min. The first observation of weight loss is related to dehydration of water molecules and decomposition of organic compounds, and the second part is attributed to decomposition of functional groups. The final weight loss is associated with the decomposition of the pyrochlore phase. The maximum weight loss of pure k-carrageenan was found to occur in two temperature ranges, 79.5–125.4 °C and 239.2–244.1 °C. The first stage is associated with water loss. The second step occurred between 239.2 and 244.1 °C, with a maximum decomposition of 21.01%. The TGA traces of Ca/PVA hydrogels were between PVA and k-carrageenan, with two separate breakdown phases. The findings of this investigation demonstrated that adding more PVA to k-carrageenan improved the thermal stability of the hydrogels. The highest breakdown rate (for Ca/PVA 70:30) was obtained at 210.3 °C and 225.6 °C, and the second stage of hydrogel decomposition demonstrated the great thermal stability of k-carrageenan hydrogels in combination with PVA. When compared to Ca/PVA 60:40, the first degradation weight loss of Ca/PVA occurred with a slower slope from 230.0 °C to 233.8 °C and the second was from 507.9 °C to 599.7 °C, indicating that the Ca/PVA 50:50 hydrogel has a great heat stability according to the first and second degradation temperature (from 231.4 °C to 234.8 °C in the first degradation and from 399.9 °C to 460.3 °C in the second degradation). This could be attributed to the presence of PVA at the same concentration as k-carrageenan in the hydrogel matrix.

Therefore, after characterizing these polymers, Ca/PVA 50:50 was chosen as the main polymer in this study, and was mixed with nanoparticles (MgO, MgZnO 1%, MgZnO 3%, and MgZnO 5%) and loaded with catechins (drug model) for further characterization.

### 3.3. SEM of Nanocomposites

Figure 2 shows the surface morphology of the nanocomposites examined by SEM. SEM images revealed that the composition of the matrix surface was affected by the composite microstructure. According to Figure 2a, a smooth surface of the k-carrageenan/PVA/MgO composite was obtained. The introduction of ZnO nanoparticles into MgO in the composites changed the surface morphology of the composites, resulting in uneven surfaces with warped sections (Figure 2b–d). As shown in Figure 2, increasing the ZnO content in the nanoparticles changed the surface of the composite to a rough surface. The surface morphologies of pure nanoparticles (MgO, MgZnO 1%, MgZnO 3%, and MgZnO 5%) and nanoparticles of PVA hydrogels have been shown in recently published data [20]. According to previous studies on the structure of nanoparticles, filling zinc oxide particles with magnesium oxide results in the formation of hexagonal shapes. In addition, by combining MgO and ZnO, the crystal formation in the direction was suppressed. Therefore, ‘Zn’ doping of magnesium oxide has a significant impact on the structure and morphological organization of MgO nanoparticles [22].

### 3.4. Texture Analysis of Nanocomposites

The dispersion of nanoparticles through matrices is affected by their mechanical properties. The results of the compression tests (consistency, adhesiveness, springiness, and hardness) of the nanocomposites are shown in Table 3. The hardness of k-carrageenan/PVA/MgZnO 5% nanocomposite was 549.926 (g) which was higher than the other nanocomposites. By increasing the ZnO content in the structure of the nanoparticles, the hardness of the nanocomposites also seems to increase accordingly. The results of springiness tests showed that the k-carrageenan/PVA/MgZnO 1% and k-carrageenan/PVA/MgZnO 3% nanocomposites were the same, at 0.201 mm, but the MgO nanocomposites were similar to the other composites. They showed a lower springiness (0.199 mm) than the other nanocomposites. However, the springiness of the k-carrageenan/PVA/MgZnO 5% nanocomposite was shown to be higher than the other nanocomposites. The results of adhesiveness indicated that increasing the ZnO ratio also increased the adhesiveness, and there is a direct relation between ZnO and adhesiveness. As shown in Table 3, the lowest adhesiveness belongs to the k-carrageenan/PVA/MgO nanocomposite (129.326 g/s) and the highest adhesiveness belongs to the k-carrageenan/PVA/MgZnO 5% nanocomposite (501.596 g/s). The consistency in the k-carrageenan/PVA/MgZnO 5% was the highest (403.248 g/s). Another study has also shown the same results for PVA polymer and MgZnO nanoparticles [22].

### 3.5. Drug Loading Capacity of Nanocomposites

The amount of drug loaded into the nanocomposite structures was also measured and isshown in Table 4. The results show that the maximum catechin drug content in the nanocomposites is 27.93%. This is a reasonable value and demonstrates Ca/PVA nanocomposites as high-loading drug carriers. In addition, since the drug is trapped in the three-dimensional structure of the nanocomposite, the drug cannot be rapidly released from the nanocomposite, and the produced nanocomposite can release the catechin as a sustained drug release. The entrapment efficiency is one of the most important factors influenced by drug properties and hydrogel fabrication methods. Therefore, the entrapment efficiency (EE%) of catechins in Ca/PVA hydrogels was determined. As shown in Table 4, the EE% of catechins in the nanocomposites ranged from 55.21% to 81.58%. This could be attributed to a possible solubilizing effect of the catechin polymer in the external aqueous phase, resulting in an increased un-trapped “free form” in the supernatant, and a consequent reduction in EE%. In a study conducted by Panwar et al. in 2010, albendazole-encapsulated liposomes were employed to provide sink conditions. The reproducibility and efficacy of the release study were ensured through a control sample containing the drug in the free form. In the case of free albendazole, more than 80% of the drug was released within the first sampling time (30 min) while both liposomal formulations produced an initial slower effect in which albendazole release was more than 25% for PEGylated and 35% for non-PEGylated liposomes within the first sampling time (30 min).

Overall, the high EE% values in all the prepared nanocomposite formulations could be attributed to the hydrophilicity of the loaded drug, which allows it to be incorporated within the polymers.

### 3.6. In Vitro Drug Release Analysis

The surface area of a drug carrier is one of the most important parameters affecting drug release behavior. The interfacial interactions between nanoparticles, k-carrageenan, and PVA can affect the surface morphology of k-carrageenan/PVA, which directly affects the drug release profile. As can be observed in Figure 2, the k-carrageenan/PVA/MgO nanocomposites exhibit a randomly and continuously ordered morphology without pores. Furthermore, no nanoparticles were observed on the surface of the nanocomposites by SEM, indicating that the nanoparticles were completely embedded in the k-carrageenan/PVA nanocomposites. Accordingly, the addition of ZnO nanoparticles to the k-carrageenan/PVA nanocomposites has obvious effects on their morphological properties. The release rate of catechins from each nanocomposite was determined. Figure 3 shows the release profile of the catechin-loaded nanocomposites. A total of 28% of the drug was released into the dissolution medium after about 2 h, reaching 75% after 12 h and remaining constant for 24 h in a sustained release. In k-carrageenan/PVA/MgZnO 5%, 75% is released after 12 h and stays constant in a sustained release for 24 h. The initial release is attributed to the migration of catechin molecules near the surface of the Ca/PVA nanocomposites, thereby inducing their rapid diffusion from the nanocomposite’s spongy surface into the buffer solution during the initial incubation period. Moreover, there is a significant difference in drug release rate compared to hydrogels without ZnO (ANOVA, *p* < 0.05). These hydrogels show a low release of 3% after 2 h, and a total release of 55% after 12 h. This is showing the effect of ZnO on the release rate of catechins from Ca/PVA nanocomposites. According to Table 4, the drug loading and entrapment efficiency results show that the MgO containing nanocomposites have a drug loading of 17.45% and an encapsulation efficiency of 55.21% compared to other nanocomposites, and this results in a reduction in the drug release. As shown in Table 4, the highest drug loading rate was 27.93% and the entrapment efficiency was 81.58%, which is related to the nanocomposite with 5% ZnO in the structure. As can be seen in Figure 3, the release profiles of catechins from the Ca/PVA nanocomposites are almost identical due to the incorporation of 1%, 3%, and 5% ZnO in the nanoparticle structure (65.55% ± 0.76, 70.31% ± 0.43, and 75.21% ± 0.94, respectively). In particular, a comparison of the catechin-loaded Ca/PVA nanocomposites shows that the addition of 5% ZnO to the Ca/PVA composites can provide a better cumulative catechin release profile than other nanocomposites. This may be related to the role of ZnO nanoparticles in Ca/PVA nanocomposites, resulting in a higher strength k-carrageenan and PVA chain. It may also be due to the small size and large surface area of ZnO, the drug nanoparticles exhibiting increased solubility, an enhancement in the bioavailability, and an increase in the release rate of catechins from ZnO nanocomposites. Therefore, the catechin-loaded Ca/PVA nanocomposites can be introduced as suitable drug delivery systems.

### 3.7. In Vitro Cytotoxicity Test Cell

Cell viability tests are essential to assess the in vitro cytotoxicity of hydrogel films for topical application. Using the MTT assay, ZnMgO-doped hydrogel films were cultured with a fixed density of mouse fibroblast L929 cell lines and individually incubated for 24, 48, and 72 h, the results of which are shown in Figure 4. Significant proliferation of fibroblast cells indicated that the hydrogel films were biocompatible. After 24 h of incubation, the viability of L929 cells was not inhibited by the presence of hydrogel films, and cell viability was higher than that of the control, especially for k-carrageenan/PVA/ZnMgO 5% hydrogel films, which showed more than 100% cell viability. A similar trend was observed after a 48-h incubation, with more than 85% cell viability. After 72 h of incubation, the viability of L929 cells declined to 76%, 85%, and 95% with k-carrageenan/PVA/ZnMgO 3% and k-carrageenan/PVA/ZnMgO 1% films, respectively. The decrease in cell viability after 72 h of incubation may be due to the nutritional deficiencies resulting from partial depletion of nutrients in the culture medium by rapidly growing L929 cells. The k-carrageenan/PVA/ZnMgO 5% hydrogel films showed significantly higher cell viability compared to the k-carrageenan/PVA/ZnMgO 3% hydrogel film.

## 4. Conclusions

Ca/PVA wound healing nanocomposites were synthesized by combining MgZnO nanoparticles, k-carrageenan, and PVA. The purpose of mixing these two polymers was to increase the stability of k-carrageenan. The structure of the nanocomposites was characterized by FTIR, TGA, and XRD. As a result of characterization, Ca/PVA 50:50 was found to be the most suitable and was used in the next step. Then, Ca/PVA 50:50 was mixed with different ratios of MgZnO nanoparticles and examined by texture analysis, SEM, and cytotoxicity studies. The results of drug release and cytotoxicity studies indicated that k-carrageenan/PVA/MgZnO 5% nanocomposites could be a suitable candidate for drug delivery applications in wound healing dressing materials with the highest rate of drug release and significant cell viability compared to the other nanocomposites.

## Figures and Tables

**Figure 1 polymers-15-00272-f001:**
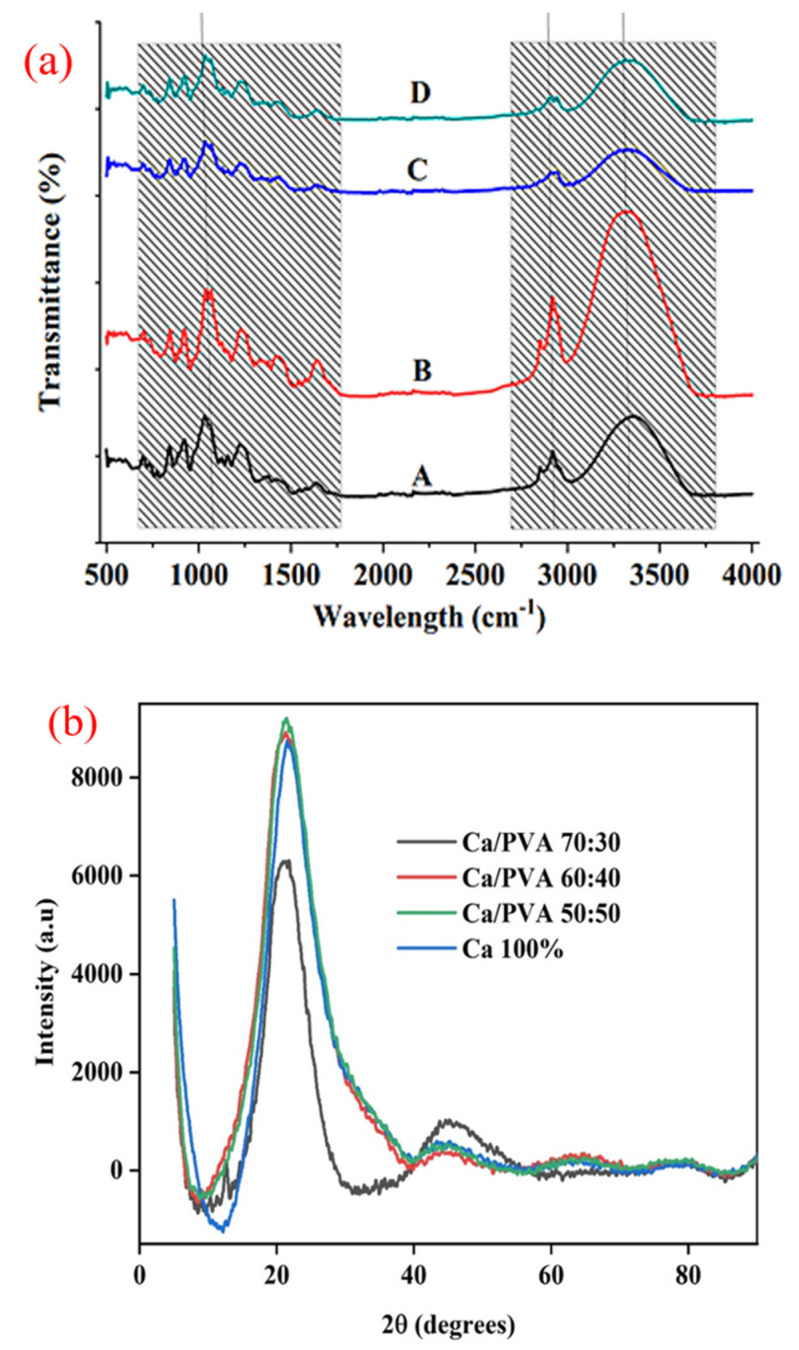
(**a**) FTIR spectra of (A) Ca/PVA 50:50, (B) Ca/PVA 60:40, (C) Ca/PVA 70:30, and (D) Ca. (**b**) XRD diffraction patterns of Ca/P composites.

**Figure 2 polymers-15-00272-f002:**
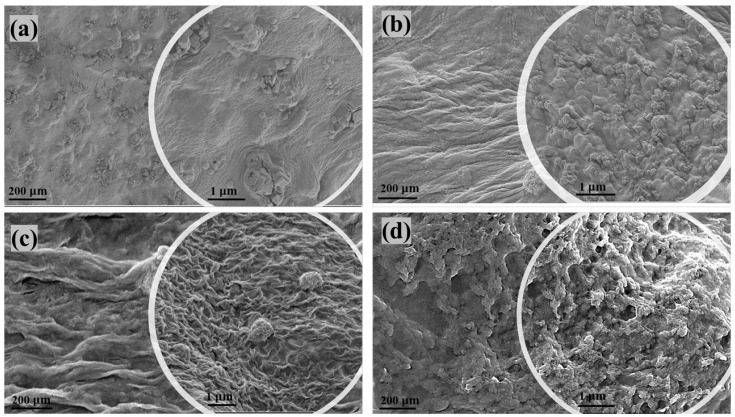
SEM images of k-carrageenan/PVA hydrogels containing MgZnO nanoparticles. (**a**) k -carrageenan/PVA/MgO, (**b**) k -carrageenan/PVA/MgZnO 1%, (**c**) k -carrageenan/PVA/MgZnO 3%, and (**d**) k -carrageenan/PVA/MgZnO 5%.

**Figure 3 polymers-15-00272-f003:**
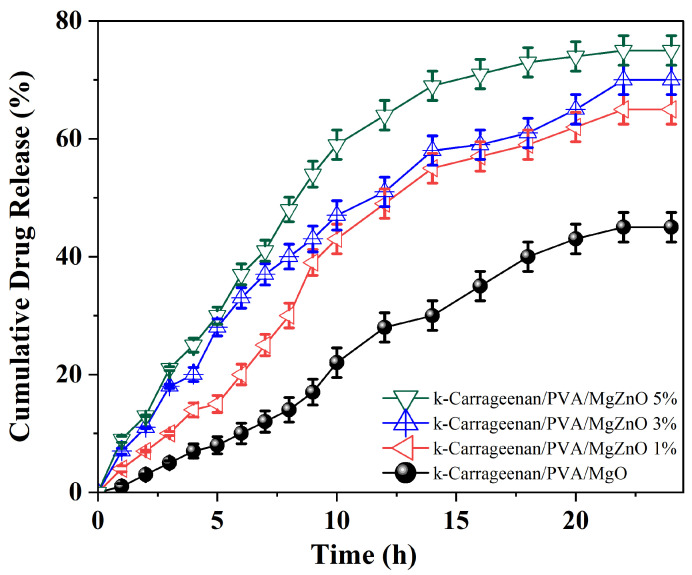
The in vitro release profile of catechin from Ca/PVA nanocomposites with different ZnO ratios. Data represent mean ± SD (*n* = 3).

**Figure 4 polymers-15-00272-f004:**
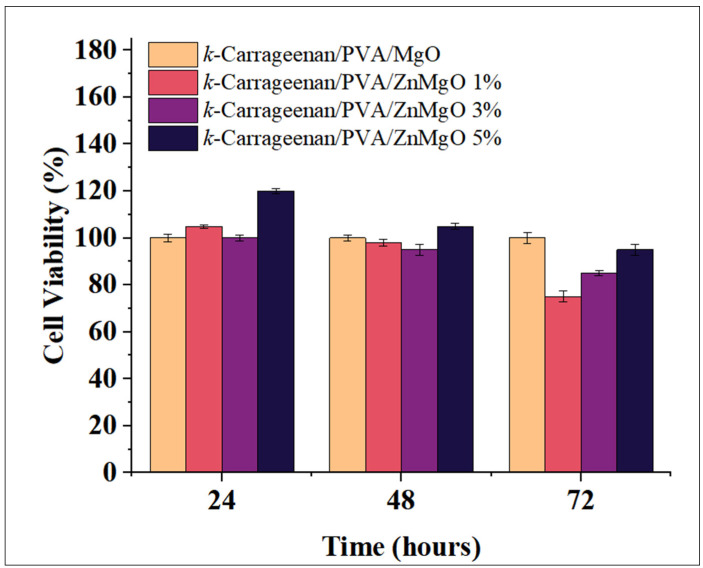
Cellular proliferation analysis (MTT) of k-carrageenan/PVA hydrogel films with different nanoparticle ratios against L929 cells.

**Table 1 polymers-15-00272-t001:** The formulation of k-Carrageenan/PVA nanocomposite composites.

	k-Carrageenan (g)	PVA (g)	Mixture (%)
(A) Ca/P	1	1	50:50
(B) Ca/P	1.2	0.8	60:40
(C) Ca/P	1.4	0.6	70:30
(D) Ca	2	-	100

**Table 2 polymers-15-00272-t002:** The details of the TGA diagrams for Ca/PVA 50:50, Ca/PVA 60:40, Ca/PVA 70:30, and Ca hydrogels.

Compound	First Degradation °C		Second Degradation °C		Mass Change %	
	Onset	End	Onset	End	First	Second
Ca/PVA 50:50	231.4	234.8	399.9	460.3	19.52	17.33
Ca/PVA 60:40	230.0	233.8	507.9	599.7	20.69	0.35
Ca/PVA 70:30	210.3	225.6	210.3	244.4	12.42	30.20
Ca	79.5	125.4	239.2	244.1	1.38	21.01

**Table 3 polymers-15-00272-t003:** Compression test of k -carrageenan/PVA hydrogels containing MgZnO nanoparticles.

Sample	Adhesiveness (g/s)	Springiness (mm)	Consistency (g/s)	Hardness (g)
k-carrageenan/PVA/MgO	129.326	0.199	56.334	129.326
k-carrageenan/PVA/MgZnO 1%	169.462	0.201	154.01	387.401
k-carrageenan/PVA/MgZnO 3%	322.529	0.201	169.22	403.248
k-carrageenan/PVA/MgZnO 5%	501.596	0.401	403.248	549.926

**Table 4 polymers-15-00272-t004:** The drug loading and entrapment efficiency (%) in k -carrageenan/PVA hydrogels containing MgZnO nanoparticles.

Nanocomposites	Drug Loading (mg/100 mg Nanocomposite)	Entrapment Efficiency (%)
k-carrageenan/PVA/MgO	17.45	55.21
k-carrageenan/PVA/MgZnO 1%	23.04	68.30
k-carrageenan/PVA/MgZnO 3%	25.55	77.42
k-carrageenan/PVA/MgZnO 5%	27.93	81.58

## Data Availability

Not applicable.

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
