# Peer review of "Synthesis and Characterization of 𝒌-Carrageenan/PVA Nanocomposite Hydrogels in Combination with MgZnO Nanoparticles to Evaluate the Catechin Release"

_polymers, 2023, doi:10.3390/polym15020272_

Round 1

Reviewer 1 Report

Review of the manuscript which has been submitted to Polymers-Manuscript no. polymers-2066623

In the current context of the study topic, the article entitled “Synthesis and characterization of ?-carrageenan/PVA nanocomposite hydrogels in combination with MgZnO nanoparticles to evaluate the Catechin release” is very interesting.  Anyway, below I have made some remarks in order to improve the quality of the paper before acceptance for publication.

·         Page 3, line 98; Please add more explanations to section 2.2. so that anyone can reproduce these techniques;

·         Page 3, line 107; Please add more explanations to the section 2.3. so that anyone can reproduce these techniques;

·         Page 3, line 121; Please write “They” with lowercase;

·         Page 4, line 138; Please reformulate the sentence “The catechin loaded nanocomposites were then immersed in PBS of same composition.” for a better understanding;

·         Page 4, line 147; Please reformulate the sentence “Structure analysis FT-IR spectra of ?-Carrageenan/PVA with different formulations” for a better understanding;

·         Page 4, line 150; please be more specific in the sentence “The presence of CH2 groups is responsible for the peak at 2940 cm-1”, the CH2 group of which structure. The observation is valid for the entire paragraph; references must be added for the composite structures relative to the existing literature data.

·         Page 5, lines 177-179; Please reformulate the sentence “The first weight 177 loss observation related to water molecules’ dehydration and breakdown of organic com-178 pounds the second part happened is attributed to the decomposing of the functional 179 groups.” for a better understanding;

·         Page 5, line 138; please reformulate correctly in concordance with Table 2 the sentences “The low heat resistance of ?-carrageenan was attributed to its great thermal stability. It revealed that most of the weight loss of neat ?-carrageenan occurred  in a single temperature range of 79.5 to 125.4 °C. The first stage is connected with water  loss. The second stage occurred between 239.2 and 244.1 °C, with a maximum decompo-184 sition rate of 21.1%.”

·         Page 5, line 186; you say “with three separate breakdown phases.” But in Table 2 there are only 2…please correct;

·         Page 5, line 194; add the TGA recordings for all samples in order to see more clearly the observations that are made;

·         Page 5, line 196; it is not correct to say the first, and second observations, this are the degradations steps…please correct.

·         Page 5, line 206; what figure? Be clear please.

·         Page 6, lines 233, 234; use a capital letter at the beginning of the sentence;

·         Page 7, line 243; Please reformulate “to the as prepare nano composite” for a better understanding;

·         Page 7, line 248; add a literature study for the observation “possible solubilizing effect of polymer for catechin in the external aqueous phase,"

·         Page 7, line 250; I see 80% EE only for ?-Carrageenan/PVA/ MgZnO 5% and you say something else, please correct!

·         Page 8, line 298; please use the same typeface for “?-carrageenan”
throughout the document.

·         Please restructure the Conclusions section, highlighting the results obtained.

Author Response

Thanks to the reviewers for helping to increase the quality of the manuscript.

Reviewer 2 Report

On the whole, the study is basically complete, and the results seem to support their claims. However, the materials are not obviously innovative, and additional points of clarifications could potentially be addressed to further strengthen the manuscript.

Comments:

1.     In line 250, the author said “Overall, the EE% values exceeding 80% in all the prepared nanocomposites formulations, which could be attributed to the hydrophilicity of the loaded drug, which allows it to be incorporated within the polymers.”, but in table 4, only EE of ZnO 5% is more than 80%, how you draw this conclusion?

2.     In line 272, the author stated that “there is a significant difference in the drug release rate concerning the hydrogels that are not containing ZnO.” I would recommend the author performed the statistical analysis and then draw the conclusion says if there is any significant differences.

3.     In line 283, the author said “As is apparent in Figure 3, there is a significant reduction in the initial burst release of catechin from the Ca/PVA nanocomposites by the incorporation of (1%) and (3%) ZnO in the structure of nanoparticles.”, but the figures showed that when you increased the percentage of ZnO, the cumulative drug release percentage is increased. The results seems there is an acceleration of the drug release and there isn’t any reduction of the burst release amount. Can you explain this?

4.     In Figure 3, please perform statistical analysis, then you can say if there is any significant reduction in line 281.

5.     In Figure 3, the catechin was only release in 12 h across all the group and then reached plateau. This is just the amount release is increased by incorporating of ZnO, but doesn’t mean the release rate has been changed. Please comment this.

6.     What is the advantage of this type of hydrogel compared to other commercial available hydrogel?

7.     The author said that this material can be used as wound healing materials, but biocompatibility is important property of biomaterials. I would recommend author add some cell toxicity studies.  

Author Response

(The authors gave the same response as above.)

Round 2

Reviewer 1 Report

The authors have made all suggested changes. Moreover, they almost completely rewrote the material. In the current situation, I agree with the article publication.

Reviewer 2 Report

All the comments were addressed properly.